# SPEECHTOKENIZER: UNIFIED SPEECH TOKENIZER FOR SPEECH LANGUAGE MODELS

**Xin Zhang**[*], **Dong Zhang**[*], **Shimin Li**, **Yaqian Zhou**[†] **Xipeng Qiu**[†]
School of Computer Science, Fudan University
Shanghai Key Laboratory of Intelligent Information Processing, Fudan University
`{xin_zhang22,dongzhang22}@m.fudan.edu.cn`
`{smli20,zhouyaqian,xpqiu}@fudan.edu.cn`

`https://0nutation.github.io/SpeechTokenizer.github.io/`

## ABSTRACT

Current speech large language models build upon discrete speech representations, which can be categorized into semantic tokens and acoustic tokens. However, existing speech tokens are not specifically designed for speech language modeling. To assess the suitability of speech tokens for building speech language models, we established the first benchmark, SLMTokBench. Our results indicate that neither semantic nor acoustic tokens are ideal for this purpose. Therefore, we propose SpeechTokenizer, a unified speech tokenizer for speech large language models. SpeechTokenizer adopts the Encoder-Decoder architecture with residual vector quantization (RVQ). Unifying semantic and acoustic tokens, SpeechTokenizer disentangles different aspects of speech information hierarchically across different RVQ layers. Furthermore, We construct a **U**nified **S**peech **L**anguage **M**odel (USLM) leveraging SpeechTokenizer. Experiments show that SpeechTokenizer performs comparably to EnCodec in speech reconstruction and demonstrates strong performance on the SLMTokBench benchmark. Also, USLM outperforms VALL-E in zero-shot Text-to-Speech tasks. Code and models are available at `https://github.com/ZhangXInFD/SpeechTokenizer/`.

## 1 INTRODUCTION

Large language models (OpenAI, 2023; Touvron et al., 2023) have demonstrated remarkable performance on various natural language processing tasks. This has inspired numerous works to build speech language models (Borsos et al., 2022), which have achieved significant breakthroughs across various speech processing tasks (Wang et al., 2023; Zhang et al., 2023; Rubenstein et al., 2023; Dong et al., 2023). A key commonality among these works is the utilization of discrete speech representations. Current discrete speech representations can be categorized into two types: semantic tokens and acoustic tokens (Borsos et al., 2022). Semantic tokens are typically from self-supervised pre-trained models with masked language modeling as training objective (Hsu et al., 2021; Baevski et al., 2020; Chung et al., 2021). Derived through k-means clustering on representations from a specific intermediate layer, semantic tokens are depicted as sequences with one-dimensional structure. Acoustic tokens can be extracted from neural audio codecs with reconstruction as training objective (Zeghidour et al., 2021; Défossez et al., 2022). Utilizing residual vector quantization (RVQ) with hierarchical quantizers for discretization, acoustic tokens are represented as matrices consisting of two dimensions: timesteps and quantizers.

Building upon two speech tokens, there exist three modeling approaches for speech language models, as listed in Table 1: *i) Semantic language models* are constructed using semantic tokens and employ an external unit vocoder for speech synthesis. (Lakhotia et al., 2021; Zhang et al., 2023; Hassid et al., 2023). While capturing semantically accurate content, their speech generation results in poor

---

[*]Equal contribution. Order is random
[†]Corresponding author

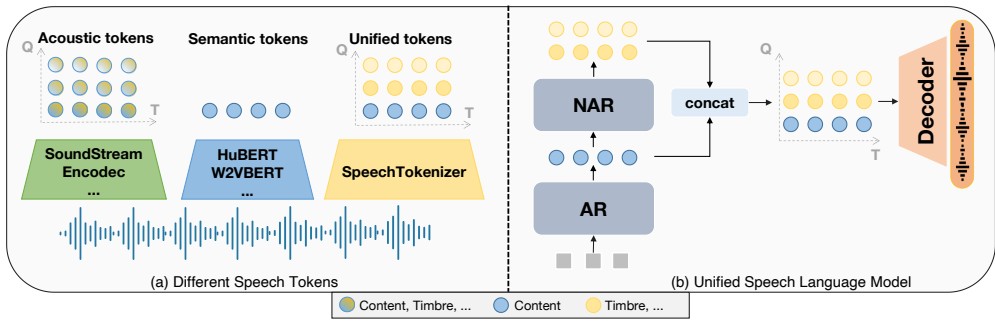

Figure 1: **Left**: Illustration of information composition of different discrete speech representations. **Right**: Illustration of unified speech language models. AR refers to autoregressive and NAR refers to non-autoregressive. Speech tokens are represented as colored circles and different colors represent different information.

| | Accurate Content | High-quality Speech | Single Tokenzier |
|---|:---:|:---:|:---:|
| Semantic LM | √ | × | √ |
| Acoustic LM | × | √ | √ |
| Hierarchical LM | √ | √ | × |
| USLM (ours) | √ | √ | √ |

Table 1: Comparision between different speech language models. *Semantic LM* refers to semantic language models. *Acoustic LM* refers to acoustic language models. *Hierarchical LM* refers to hierarchical speech language models. *USLM* refers to our unified speech language model.

quality and a loss of acoustic details. *ii) Acoustic language models* are built on acoustic tokens. Taking VALL-E (Wang et al., 2023) as an example, despite achieving impressive zero-shot text-to-speech (TTS) capabilities, it still suffers from problems like inaccurate content, due to the complex information within acoustic tokens. *iii) Hierarchical speech language models* comprise semantic token language models and acoustic token language models, which capture content information and acoustic details respectively (Borsos et al., 2022; Rubenstein et al., 2023; Dong et al., 2023). This structure shows promising results in both content and speech quality, but the multi-stage modeling approach is more complex, leading to several drawbacks such as error accumulation and slower processing speed. Additionally, there is significant information redundancy between semantic tokens and acoustic tokens, which introduces unnecessary modeling complexities. An ideal speech language model should not only accurately model content, but also generating diverse, high-quality speech, while maintaining an architecture of elegant simplicity. Correspondingly, ideal speech tokens should meet the following two key characteristics: i) Strong alignment with text; ii) Effective preservation of speech information.

However, existing speech tokens are not explicitly designed for speech language modeling, and there has been no exploration into their suitability for building speech language models. To address this gap, we build Speech Language Model Token Benchmark, to assess the suitability of speech tokens for constructing speech language models. Our evaluation reveals that semantic tokens exhibit a high alignment with text while losing some information in speech, such as timbre. Acoustic tokens excel in preserving speech information effectively but do not demonstrate a strong alignment with text. With these observations, we aim to build a specialized speech tokens designed for speech language models by unifying semantic and acoustic tokens. Specifically, we can conduct information disentanglement in the RVQ structure of acoustic tokens, enabling the first RVQ quantizer to generate tokens containing content information, similar to semantic tokens, while the subsequent quantizers complement the remaining paralinguistic information, as illustrated in Figure 1.

With the above motivation, we propose SpeechTokenizer, a unified speech tokenizer for speech large language models. SpeechTokenizer adopts the Encoder-Decoder architecture with residual vector quantization. Unifying semantic and acoustic tokens, SpeechTokenizer disentangles different aspects of speech information hierarchically across different RVQ layers. By employing a semantic teacher to guide the first RVQ quantizer, the first layer tokens can effectively capture content information. With residual structure, the subsequent quantizers complement the remaining paralinguistic information.

Building upon SpeechTokenizer, we build a **U**nified **S**peech **L**anguage **M**odel consisting of autoregressive and non-autoregressive models. Experimental results show that SpeechTokenizer performs comparably to EnCodec (Défossez et al., 2022) in speech reconstruction and demonstrates strong performance on the SLMTokBench benchmark. The USLM notably outperforms VALL-E (Wang et al., 2023) in zero-shot Text-to-Speech (TTS) tasks.

Our contributions include the following:

- We propose SpeechTokenizer, which is specially designed for speech large language models and unify the semantic and acoustic tokens through disentangling different aspects of speech information hierarchically.
- We establish SLMTokBench, the first benckmark to assess the suitability of speech tokens for constructing speech language models.
- We construct a unified speech language model based on SpeechTokenizer, which outperforms VALL-E on zero-shot TTS task.

## 2    SLMTokBench: Speech Language Model Token Benchmark

To build powerful speech language models, discrete speech representations should possess the following two key characteristics: i) Strong alignment with text; ii) Effective preservation of speech information. Building on this premise, we establish speech Language Model Token Benchmark (SLM-TokBench) to assess the suitability of speech tokens for constructing speech language models.

### 2.1    Text Alignment Evaluation

We evaluate the degree of text alignment by estimating the mutual information between speech tokens and text. For notation, $\mathbf{X}$ denotes discrete speech representations; $\mathbf{Y}$ denotes text; $\mathcal{I}(\mathbf{X}; \mathbf{Y})$ denotes the mutual information; test dataset is denoted as $\mathcal{D} = \{(x_i, y_i)\}_{i=1}^{N}$ and $\theta$ denotes the downstream model. Through the derivation in Appendix A, we can estimate $\mathcal{I}(\mathbf{X}; \mathbf{Y})$ as:

$$\hat{\mathcal{I}}(\mathbf{X}; \mathbf{Y}) = \frac{1}{N^2} \sum_{i=1}^{N} \sum_{j=1}^{N} [\log q_\theta(y_i|x_i) - \log q_\theta(y_j|x_i)]$$

where $q_\theta(\mathbf{Y}|\mathbf{X})$ is the variational distribution and can be parameterized by the downstream model $\theta$.

The downstream model is a vanilla 2-layer 1024-unit BLSTM optimized by CTC loss on characters and it takes speech tokens as inputs. Specifically, for each discrete representation, we first establish an embedding matrix, which can be either randomly initialized or derived from the k-means centroid matrix or vector quantization codebooks obtained during the discretization process. We use the embedding matrix to embed the discrete representations and obtain continuous representations, which are then fed into the downstream models. We train the downstream model on LibriSpeech train-clean-100 subset and use dev-clean subset for estimating mutual information. We also calculate the word error rate (WER) on the test set. For downstream model training, we configure the training setup with a batch size of 32, a learning rate of 1e-4, and a total of 200k global steps.

### 2.2    Information Preservation Evaluation

To evaluate the preservation of speech information in discrete speech representations, we convert speech tokens back to speech and evaluate resynthesized speech by automatic metrics on content and timbre. We train a unit-HiFIGAN (Polyak et al., 2021) on LibriSpeech dataset to convert HuBERT units to waveform. Notably, to avoid interference from additional information, we don't supply any speaker information during training. For Encodec tokens, we used the Encodec decoder to directly produce the waveform. Content preservation is evaluated by computing the WER through transcribing the resynthesized speech using the Whisper en-medium model (Radford et al., 2023). Timbre preservation is evaluated by utilizing WavLM-TDNN (Chen et al., 2022) to calculate speaker similarity between the synthesized and groundtruth speech. We randomly sample 300 speech samples from LibriSpeech test set for evaluation.

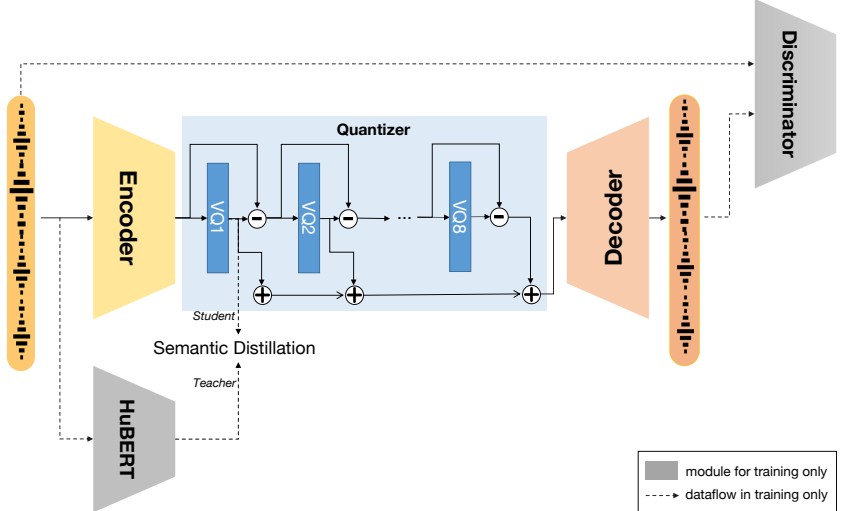

Figure 2: Illustration of SpeechTokenizer framework.

## 2.3 COMPARING SEMANTIC & ACOUSTIC TOKENS

We use HuBERT L9 units to represent semantic tokens and EnCodec codes to represent acoustic tokens. As shown in Table 3, semantic tokens achieve high mutual information with text but their resynthesized speech has low speaker similarity. Acoustic tokens achieve low WER and high speaker similarity for resynthesized speech but have low mutual information with text.

## 3 SPEECHTOKENIZER

### 3.1 MODEL STRUCTURE

Our model is built on the framework of RVQ-GANs, following the same pattern as Sound-Stream(Zeghidour et al., 2021) and EnCodec(Défossez et al., 2022). As depicted in Figure2, our model uses the convolutional-based encoder-decoder network from EnCodec, which performs temporal downscaling with a chosen striding factor. Notably, we have substituted the two-layer LSTM, originally following the convolution blocks in the EnCodec encoder, with a two-layer BiLSTM to augment the semantic modeling ability. We conduct ablation studies of model structure in Appendix B. We quantize the encoder outputs using Residual Vector Quantization (RVQ), a method that can operate quantizes residuals following an initial quantization steps with distinct codebook. Further details of model structure can be found in Appendix D. During training, a semantic teacher provides semantic representation to guide the residual quantization process.

### 3.2 SEMANTIC DISTILLATION

To achieve a hierarchical modeling of diverse information across different RVQ layers, we employ semantic guidance for the first quantizer, enabling it to capture content information. Leveraging a residual structure enables the subsequent quantizers to complement the remaining paralinguistic information.

We employ HuBERT (Hsu et al., 2021) as our semantic teacher in this study, as HuBERT is demonstrated to encompass substantial content information (Mohamed et al., 2022). We introduce two types of distillation: continuous representation distillation and pseudo-label prediction.

For continuous representation distillation, we employ the 9th layer HuBERT representation or the average representation across all HuBERT layers as semantic teachers. The training objective is to maximize the cosine similarity at the dimension level across all timesteps between the outputs of RVQ first layer and semantic teacher representations. Formally, the continuous distillation loss is

defined as:

$$\mathcal{L}_{distill} = -\frac{1}{D} \sum_{d=1}^{D} \log \sigma(cos(\mathbf{A}\mathbf{Q}_1^{(:,d)}, \mathbf{S}^{(:,d)})),$$

where $\mathbf{Q}_1$ and $\mathbf{S}$ denote the quantized output of RVQ first layer and semantic teacher representation respectively. $\mathbf{A}$ denotes the projection matrix and $D$ is the dimension of semantic teacher representation. The superscript $(:, d)$ signifies a vector comprising values from all timesteps at dimension $d$. $cos(\cdot)$ represents cosine similarity and $\sigma(\cdot)$ denotes sigmoid activation. This continuous distillation loss function deviates from the commonly employed approach, which calculates the loss based on the representations output by the student and teacher models at the same timestep. A comparative analysis of these two methodologies is provided in Appendix C.

For pseudo-label prediction, we adopt HuBERT units as the target label. The training objective is constructed as:

$$\mathcal{L}_{distll} = -\frac{1}{T} \sum_{t=1}^{T} \mathbf{u}^t \log(\text{Softmax}(\mathbf{A}\mathbf{q}_1^t)),$$

where $\mathbf{q}_1^t$ and $\mathbf{u}^t$ respectively denote the quantized output of the first VQ layer and the HuBERT unit at timestep t. $T$ denotes the number of time steps and $\mathbf{A}$ is the projection matrix.

### 3.3 TRAINING OBJECTIVE

Our training approach includes both a reconstruction task and a semantic distillation task. In the reconstruction task, we employ a GAN objective, optimizing a combination of a reconstruction term, a discriminative loss term, and RVQ commitment loss. In the semantic distillation task, the training objective involves a semantic distillation loss term. In the following, $\mathbf{x}$ represents an speech signal and $\hat{\mathbf{x}}$ denotes the reconstructed signal by the network.

**Reconstruction Loss** The reconstruction loss comprises a time and a frequency domain loss. For time domain, we minimize the L1 distance between $x$ and $\hat{x}$, i.e. $\mathcal{L}_t = \|\mathbf{x} - \hat{\mathbf{x}}\|_1$. For frequency domain, we linearly combine the L1 and L2 losses over the mel-spectrogram using several time scales. Formally, $\mathcal{L}_f = \sum_{i \in e}\|\mathcal{S}_i(\mathbf{x}) - \mathcal{S}_i(\hat{\mathbf{x}})\|_1 + \|\mathcal{S}_i(\mathbf{x}) - \mathcal{S}_i(\hat{\mathbf{x}})\|_2$, where $\mathcal{S}_i$ is a 64-bins mel-spectrogram using a normalized STFT with window size of $2^i$ and hop length of $2^i/4, e = 5, \cdots, 11$ is the set of scales.

**Discriminative Loss** We use the same dicriminators as HiFi-Codec Yang et al. (2023) that consist of three discriminators: A multi-scale STFT-based (MS-STFT) discriminator; a multi-period discriminator (MPD) and a multi-scale discriminator (MSD). Further details of discriminators can be found in Appendix D. The adversarial loss is used to promote perceptual quality and it is defined as a hinge loss over the logits of the discriminator, averaged over multiple discriminators and over time. Let $K$ denote the number of discriminators, the adversarial loss for the generator $\mathcal{L}_D$ is constructed as follows, $\mathcal{L}_g = \frac{1}{K} \sum_{k=1}^{K} max(1 - D_k(\hat{\mathbf{x}}), 0)$. For the discriminators $\mathcal{L}_g$ is defined as:

$$\mathcal{L}_D = \frac{1}{K} \sum_{k=1}^{K} max(1 - D_k(\mathbf{x}), 0) + max(1 + D_k(\hat{\mathbf{x}}), 0),$$

Additionally, a feature matching loss for the generator is computed as follow:

$$\mathcal{L}_{feat} = \frac{1}{KL} \sum_{k=1}^{K} \sum_{l=1}^{L} \frac{\|D_k^l(\mathbf{x}) - D_k^l(\hat{\mathbf{x}})\|_1}{mean(\|D_k^l(\mathbf{x})\|_1)},$$

where the mean is computed over all dimensions and $L$ is the number of layers in discriminators.

**RVQ Commitment Loss** We add a commitment loss $\mathcal{L}_w$ between the pre-quantized value, and its quantized value, without gradient computed for the quantized value. RVQ commitment loss is defined as: $\mathcal{L}_w = \sum_{i=1}^{N_q}\|\mathbf{z}_i - \mathbf{z}_{q_i}\|_2^2$., where $\mathbf{z}_i$ and $\mathbf{z}_{q_i}$ denote current residual and nearest entry in the corresponding codebook respectively.

Generally, the generator is trained to optimize the following loss:

$$\mathcal{L}_G = \lambda_t \mathcal{L}_t + \lambda_f \mathcal{L}_f + \lambda_g \mathcal{L}_g + \lambda_{feat} \mathcal{L}_{feat} + \lambda_w \mathcal{L}_w + \lambda_{distill} \mathcal{L}_{distill},$$

where $\lambda_t, \lambda_f, \lambda_g, \lambda_{feat}, \lambda_w$ and $\lambda_{distill}$ are hyper-parameters used to balance each loss term.

### 3.4 Unified Speech Language Model

As shown in Figure 1, we can build a unified speech language model upon SpeechTokenizer. Consisting of autoregressive and non-autoregressive models, it can hierarchically model information in speech. The autoregressive (AR) model captures the content information by modeling tokens from the first RVQ quantizer. The non-autoregressive (NAR) model complements paralinguistic information for the AR model by generating tokens from the subsequent quantizers conditioned on the first-layer tokens. We validate the effectiveness of unified speech language model on zero-shot TTS task.

The AR model is built upon the first-layer tokens $\mathbf{c}_1$. Utilizing a transformer decoder-only architecture $\theta_{AR}$, we approach this conversion as a casual language modeling task with the phoneme sequence $u$ serving as the prompt for the AR model. The training objective can be formulated as

$$\mathcal{L}_{AR} = -\log \prod_{t=0}^{T} p(\mathbf{c}_1^t | \mathbf{c}_1^{<t}, \mathbf{u}; \theta_{AR}).$$

The NAR model produces tokens $\mathbf{c}_{2:8}$ from the subsequent quantizers. Its architecture resembles that of the AR model, comprising eight distinct acoustic embedding layers and output prediction layers. To control the characteristics of the speaker's voice, n acoustic prompt $\hat{\mathbf{C}}$ is employed for timbre guidance. The model is conditioned on phoneme sequence $u$, acoustic prompts $\hat{\mathbf{C}}$ and tokens from previous quantizers, leading to the formulation of the training objective as follows

$$\mathcal{L}_{NAR} = -\log \prod_{i=2}^{8} p(\mathbf{c}_i | \mathbf{c}_{<i}, \hat{\mathbf{C}}, \mathbf{u}; \theta_{NAR}).$$

During inference, we convert text input to phoneme sequence and speech prompt to speech tokens. They are concatenated to form the prompts for AR and NAR models. Conditioned on that, the AR model generates first-level tokens, while the NAR model iteratively produces tokens of subsequent levels. The tokens generated by the AR and NAR models are then concatenated to construct the speech token matrix. Finally, we use the SpeechTokenizer decoder to generate the waveform conditioned on the complete token matrix.

## 4 Experiments

### 4.1 Experimental Setups

**Datasets** For SpeechTokenizer training, we use LibriSpeech (Panayotov et al., 2015) dataset. We randomly crop a 3-second segment from the speech samples at each training iteration. For zero-shot TTS, we train AR and NAR models on the English subset of Multilingual LibriSpeech (Pratap et al., 2020) dataset, which contains 44K hours of transcribed speech data derived from LibriVox audiobooks. We select speech samples with durations ranging from 3 to 14 seconds for training data. The sampling rate is 16KHz for all speech data.

**Model** For SpeechTokenizer, we introduce the details about model structure in section 3.1 and Appendix D. For zero-shot TTS experiments, AR model and NAR model are both 12-layer Transformer decoders with 16 attention heads, an attention dimension of 1024 and the FFN dimension of 4096.

**Training** For SpeechTokenizer, the model are trained on 2 A800 GPUS for 20 epochs with maximum learning rate of 4e-4 and batch size of 20 per GPU. For Unified Speech Language Model, both AR and NAR models are trained on 8 A800 GPUS for 500k steps with maximum learning rate of 5e-4. The AR model is trained with batch size of 7500 tokens per GPU, and the NAR model is trained with batch size of 5000 tokens per GPU.

**Baselines** We adopt EnCodec_24khz_6kpbs (hereinafter referred to as EnCodec) (Défossez et al., 2022) as the baseline for SpeechTokenizer and VALL-E (Wang et al., 2023) as the baseline system for zero-shot TTS. We train VALL-E under the same dataset and experimental setups as EnCodec.

| Tokenizer | Objective | | Subjective |
|---|---|---|---|
| | WER↓ | VISQOL↑ | MUSHRA↑ |
| Groundtruth | 4.58 | - | 91.46 |
| EnCodec | 5.11 | 4.37 | 79.86 |
| SpeechTokenizer | 5.04 | 4.30 | 90.55 |

Table 2: Results of speech reconstruction

## 4.2 SPEECH RECONSTRUCTION EVALUATION

We randomly sample 300 speech samples from LibriSpeech test set for speech reconstruction evaluation. We take into account both subjective and objective evaluation metrics.

**Objective Metrics**  We use ViSQOL metrics (Hines et al., 2012) to measure the speech quality. Additionally, we evaluate content accuracy through Word Error Rate (WER) by transcribing the speech utilizing the Whisper en-medium model (Radford et al., 2023).

**Subjective Metrics**  We adopt a crowd-sourced methodology inpspired by MUSHRA protocol (Series, 2014), with a hidden reference but no lowerpass-filtered anchor, for subjective evaluation. We instruct evaluators to rate the perceptual quality of the given samples on a scale of 1 to 100.

## 4.3 UNIFIED SPEECH LANGUAGE MODEL EVALUATION

We conduct zero-shot TTS evaluation on the VCTK dataset, which comprises 108 speakers. There is no speaker overlap between the training data and VCTK dataset. For each speaker, we randomly selected a 3s utterance as the prompts while the textual content of a different utterance is used as the input text.

**Objective Metrics**  We evaluate the TTS systems with speaker similarity and WER. We evaluate the speaker similarity between the generated speech and the prompt speech. We calculate the similarity with the following steps: 1) we utilize WavLM-TDNN to calculate the speaker embedding for the generated speech and the prompt speech. 2) we calculate the cosine similarity between the normalized embeddings. We employ Whisper medium model to transcribe the generated speech and calculate the WER.

**Subjective Metrics**  We determine the Mean Opinion Score (MOS) and Similarity Mean Opinion Score (SMOS) through human evaluations. MOS reflects the naturalness of speech, while SMOS assesses the degree of similarity to the original speaker's voice. We engaged 12 and 6 native speakers as contributors for MOS and SMOS evaluations, respectively. MOS and SMOS both span from 1 to 5, with higher values signifying greater speech quality and voice similarity respectively.

## 4.4 MAIN RESULTS

**Speech Reconstruction**  Table 2 summarizes the results of speech reconstruction experiments. The SpeechTokenizer achienves lower WER than Encodec, demonstrating its superior ability to preserve content. Additionally, SpeechTokenizer attains a comparable VISQOL score but a higher MUSHRA score than EnCodec, which indicates its stronger capability in generating high-quality speech.

**Performance on SLMTokBench**  Table 3 displays the performance of SpeechTokenizer on SLM-TokBench. Compared with EnCodec-RVQ-1, SpeechTokenizer-RVQ-1 achieves higher mutual information between text and lower WER of downstream model. This suggests that SpeechTokenizer exhibits a stronger alignment with textual content. Meanwhile, the of resynthesized speech of Speech-Tokenizer RVQ-1 tokens achieves lower WER and speaker similarity, indicating its capability to retain more content-related information while disregarding timbre characteristics, similar to semantic tokens. The resynthesized speech of SpeechTokenizer RVQ-1:8 tokens demonstrates low WER and high speaker similarity, illustrating SpeechTokenizer's competence in preserving comprehensive speech information, similar to acoustic tokens. Furthermore, the speaker similarity of resynthesized speech of SpeechTokenizer RVQ-1 tokens is notably low, whereas that of SpeechTokenizer RVQ-1:8 tokens is considerably high. This observation implies that the tokens from subsequent layers compensate for the timbre information that is discarded by the first layer tokens.

| Tokenizer | | Teacher | Text Alignment | | Information Preservation | |
|---|---|---|---|---|---|---|
| | | | MI↑ | WER$^{\dagger}$ ↓ | WER$^{*}$ ↓ | SIM↑ |
| Groundtruth | | | - | - | 4.58 | 1.0 |
| HuBERT | KM500 | - | 31.2 | 9.88 | 16.26 | 0.77 |
| EnCodec | RVQ-1 | - | 16.5 | 61.52 | 38.34 | 0.92 |
| EnCodec | RVQ-1:8 | - | 23.6 | 30.91 | 5.11 | 0.98 |
| *Ablations* | | | | | | |
| SpeechTokenizer | RVQ-1 | HuBERT avg | 30.9 | 15.58 | 9.57 | 0.74 |
| SpeechTokenizer | RVQ-1:8 | HuBERT avg | 29.7 | 16.03 | 5.04 | 0.97 |
| SpeechTokenizer | RVQ-1 | HuBERT L9 | 32.9 | 12.68 | 14.17 | 0.73 |
| SpeechTokenizer | RVQ-1:8 | HuBERT L9 | 31.6 | 13.12 | 5.31 | 0.97 |
| SpeechTokenizer | RVQ-1 | HuBERT units | 24.2 | 34.13 | 20.02 | 0.72 |
| SpeechTokenizer | RVQ-1:8 | HuBERT units | 25.1 | 30.71 | 5.84 | 0.95 |

Table 3: Results on SLMTokBench. MI and WER$^{\dagger}$ refer to mutual information and word error rate of the downstream model. WER$^{*}$ and SIM refer to word error rate and speaker similarity of resynthesized speech respectively. RVQ-$n$ denotes the tokens of the $n^{th}$ RVQ layer. RVQ-$n$:$m$ denotes the tokens from the $n^{th}$ layer to the $m^{th}$ layer.

| Model | Tokenizer | Objective | | Subjective | |
|---|---|---|---|---|---|
| | | WER↓ | SIM↑ | MOS↑ | SMOS↑ |
| Groundtruth | | 1.9 | 0.93 | 4.5 | 3.96 |
| VALL-E | EnCodec | 7.9 | 0.75 | 3.08 | 3.31 |
| USLM | SpeechTokenizer | **6.5** | **0.84** | **3.63** | **3.45** |

Table 4: Results of zero-shot TTS

**Zero-shot TTS** As shown in Table 4, our USLM demonstrates lower WER than VALL-E. This result highlights that SpeechTokenizer can contribute to a more precise modeling of content information. Additionally, the USLM demonstrates superior speaker similarity, implying that a decoupled information structure is more conducive to modeling speaker-related information.

## 5 ANALYSIS

### 5.1 CHOICES OF SEMANTIC TEACHERS

As shown in Table 3, as semantic teachers, HuBERT L9 representations perform better than HuBERT units in both Text Alignment and Information Preservation, regardless of whether it's RVQ-1 or RVQ-1:8. The reason may be that discrete HuBERT units lose some content information compared to the continuous representations, thereby providing weaker semantic guidance to SpeechTokenizer.

When comparing HuBERT L9 representations with HuBERT average representations, we find that in terms of Text Alignment, the mutual information is higher when HuBERT L9 representations serve as the teacher. This is because HuBERT average representations contain some timbre information, while HuBERT L9 offers purer content information. On the other hand, HuBERT average shows better performance in Information Preservation, reflected in a lower WER. We speculate that this is due to a certain level of task conflict between semantic distillation and reconstruction, where the former aims to retain only content information while the later aims to preserve various aspects of speech. The presence of some timbre information in HuBERT average representations could to some extent alleviate this task conflict.

### 5.2 EFFECTIVENESS OF INFORMATION DISENTANGLEMENT

To demonstrate that different speech information can be hierarchically modeled in SpeechTokenizer, we conduct one-shot voice conversion (VC) experiment. This experiment aims to convert speech from any source speaker to an arbitrary target speaker using only a few seconds of reference speech from the target speaker. To use SpeechTokenizer for one-shot VC, the first step is to transform the source speech and reference speech into token matrices. By concatenating the RVQ-1 tokens

| Source | Reference | WER↓ | SIM↑ |
|--------|-----------|------|------|
| Groundtruth | | 0.4 | 0.93 |
| RVQ-1 | RVQ-2 | 2.6 | 0.72 |
| RVQ-1 | RVQ-2:4 | 11.7 | 0.80 |
| RVQ-1 | RVQ-2:8 | 35.4 | 0.82 |

Table 5: Results of one-shot voice conversion. Source and Reference refers to source token matrix and reference token matrix respectively.

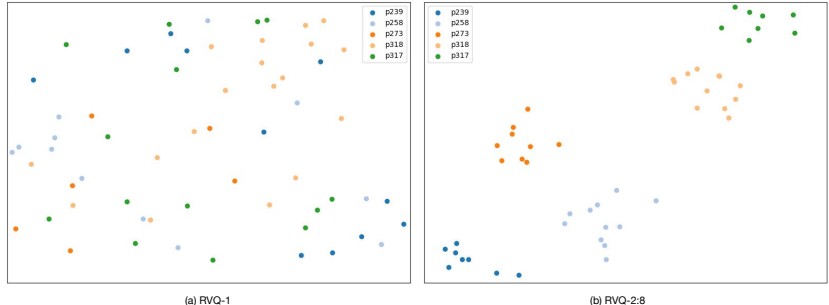

Figure 3: Visualization of quantized output of different RVQ layers of SpeechTokenizer.The first layer is denoted as RVQ-1, while the sum of the second layer to the eighth layer is denoted as RVQ-2:8.

of source token matrix with RVQ-2:8 tokens of the reference token matrix, and then passing this combined token matrix to the decoder, we can achieve voice conversion. The lengths of the reference and source tokens may not align perfectly. To address this, we use truncation or circular padding to ensure they share the same temporal length, thereby facilitating the concatenation process. We conduct experiments on VCTK dataset. We randomly selected one speech sample from a speaker to serve as the source speech. From the remaining 107 speakers, we individually selected one speech sample of different content to act as the reference speech. We employed two metrics for evaluation: WER and speaker similarity.

Table 5 reports the results of one-shot VC experiments. From the table, we can see that as the number of layers for reference tokens increases, speaker similarity also gradually increases. This suggests that more information from the reference speaker is being transferred over, proving that speaker information is embedded in tokens from the second to the last layers. When the reference tokens are selected from the second to the fourth layers, we achieve low WER and high speaker similarity, resulting in a satisfactory one-shot VC performance. This indicates that the information disentanglement is successful.

We also visualize quantized outputs from different layers in Figure 3. Specifically, We randomly select five speakers from the VCTK dataset and pick 10 random speech samples per speaker. We extract quantized output of different RVQ layers of SpeechTokenizer. The first layer output is denoted as RVQ-1 representations, while the sum of the outputs from the second layer to the eighth layer is denoted as RVQ-2:8 representations. By performing mean pooling along the temporal dimension, each representation is converted into a single vector. These vectors are then visualized in a 2D space using t-SNE, with speech samples from the same speaker represented in the same color. From the plot, it can be observed that the RVQ-1 representations for different speakers are scattered randomly without discernible pattern. In contrast, the RVQ-2:8 representations for the same speaker tend to cluster together, while being distinct from those of other speakers. This suggests that speaker-specific information is contained from the second layer up to the eighth layer.

## 6 RELATED WORK

Oure related work is put in Appendix E.

## 7 CONCLUSION

In this study, we present SLMTokBench, which assess the effects of various speech token kinds. Meanwhile, we propose SpeechTokenizer, to unify the discretization of both types of speech tokens to overcome the issue of employing several models to extract semantic and acoustic discrete tokens separately. Furthermore, We developed a unified speech language model (USLM) based on Speech-Tokenizer, with better results regarding the generated speech's content accuracy and quality. The study of a unified speech tokenizer is an essential part of the further development of speech language model in terms of efficiency and quality.

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

# A  MUTUAL INFORMATION ESTIMATION

For notation, $X$ denotes discrete speech representations; $Y$ denotes text; $\mathcal{I}(X;Y)$ denotes the mutual information; test dataset is denoted as $\mathcal{D} = \{(x_i, y_i)\}_{i=1}^{N}$ and $\theta$ denotes the downstream model. A measure of mutual information between variable $X$ and $Y$ can be formulated as:

$$\mathcal{I}(X;Y) = \int_X \int_Y log \frac{P(X,Y)}{P(X)P(Y)}$$

where $P(X)$ and $P(Y)$ are the marginal distributions of $X$ and $Y$ respectively, and $P(X, Y)$ denotes the joint distribution of X and Y.

The variational contrastive log-ratio upper bound (vCLUB) (Cheng et al., 2020) of mutual information is defined by:

$$\mathcal{I}(X;Y) = E_{p(X,Y)}[\log q_\theta(Y|X)] - E_{p(X)p(Y)}[\log q_\theta(Y|X)]$$

where $q_\theta(Y|X)$ is the variational distribution to approximate the ground-truth probability $P(Y|X)$ and can be parameterized by the downstream model $\theta$.

With test dataset $\mathcal{D}$, $\mathcal{I}(X;Y)$ has an unbiased estimation as:

$$\hat{\mathcal{I}}(X;Y) = \frac{1}{N^2} \sum_{i=1}^{N} \sum_{j=1}^{N} [\log q_\theta(y_i|x_i) - \log q_\theta(y_j|x_i)]$$

## B    MODEL STRUCTURE ABLATIONS

We conducted an ablation study on whether to use LSTM or BiLSTM. In the table 6, it can be seen that the performance of BiLSTM on text alignment is better than that of LSTM, indicating that BiLSTM is better at capturing semantic information.

| Model Structure | | Text Alignment | | Information Preservation | |
|---|---|---|---|---|---|
| | | MI↑ | WER$^\dagger$ ↓ | WER$^*$ ↓ | SIM↑ |
| CNN+LSTM | RVQ-1 | 27.60 | 20.71 | 9.06 | 0.74 |
| | RVQ-1:8 | 28.61 | 20.38 | 5.44 | 0.97 |
| CNN+BiLSTM | RVQ-1 | 30.9 | 15.58 | 9.57 | 0.74 |
| | RVQ-1:8 | 29.7 | 16.03 | 5.04 | 0.97 |

Table 6: Results of BiLSTM ablation experiment on SLMTokBench. We employ the average representation across all HuBERT layers as semantic teachers in this experiment.

## C    CONTINUOUS DISTILLATION LOSS ANALYSIS

In extant literature, the commonly employed loss functions for continuous sequence distillation are typically computed along the temporal axis, with the objective of minimizing the difference between the student and teacher model outputs at each timestep. For instance, the loss function proposed in (Chang et al., 2022) aims to maximize the cosine similarity between the student and teacher model representations at the same timestep while minimizing their $L1$ distance, thereby facilitating the transfer of knowledge from the teacher to the student model. To adapt this formula for our specific task, we can modify the the loss function as follows:

$$\mathcal{L}_{distll} = \mathcal{L}_{l1} + \lambda \mathcal{L}_{cos}$$
$$= \sum_{t=1}^{T} [\frac{1}{D} \|\mathbf{s}^t - \mathbf{A}\mathbf{q}_1^t\|_1 - \lambda \log \sigma(\cos(\mathbf{s}^t, \mathbf{A}\mathbf{q}_1^t))],$$

where $\mathbf{q}_1^t$ and $\mathbf{s}^t$ respectively denote the quantized output of RVQ first layer and the $D$ dimensional semantic teacher representation at timestep $t$. $\cos(\cdot)$ is cosine similarity. $T$ denotes the number of timesteps and $A$ is the projection matrix. $\sigma(\cdot)$ denotes sigmoid activation. $\lambda > 0$ controls the contribution of the cosine layers. We refer to this loss function as "T-axis" to distinguish it from the "D-axis" loss function that we propose in Section 3.2. The latter term is used to denote the loss function introduced in the aforementioned section. These designations are employed to differentiate between these two types of loss functions in this paper.

We investigated the impact of two distinct continuous distillation loss functions on the performance of SpeechTokenizer on SLMTokBench. The results of this experiment are summarized in Table 7. When compared to the performance of EnCodec on SLMTokBench, as presented in Table 3, employing the "T-axis" continuous distillation loss function significantly enhances SpeechTokenizer's capability in text alignment. However, this improvement is somewhat inferior to that achieved by SpeechTokenizer utilizing the "D-axis" loss function. In terms of Information Preservation, SpeechTokenizer with the "D-axis" loss function also outperforms its "T-axis" counterpart. The experimental results demonstrate that the "D-axis" continuous distillation loss function yields superior distillation effects compared to the traditional "T-axis" loss function. We attribute this improvement to the "D-axis" loss function's strategy of calculating cosine similarity across each dimension, ensuring that the student model closely aligns with the teacher model on each feature dimension. This approach provides a richer supervision signal, promoting the learning process of the student model by focusing not only on the overall output similarity but also on the similarity within each dimension.

| $\mathcal{L}_{distill}$ | | Text Alignment | | Information Preservation | |
|---|---|---|---|---|---|
| | | MI↑ | WER$^\dagger$ ↓ | WER$^*$ ↓ | SIM↑ |
| T-Axis | RVQ-1 | 26.65 | 21.10 | 10.75 | 0.76 |
| | RVQ-1:8 | 25.97 | 21.54 | 5.29 | 0.96 |
| D-Axis | RVQ-1 | 30.9 | 15.58 | 9.57 | 0.74 |
| | RVQ-1:8 | 29.7 | 16.03 | 5.04 | 0.97 |

Table 7: Results of continuous distillation loss ablation experiment on SLMTokBench. We employ the average representation across all HuBERT layers as semantic teachers in this experiment.

# D    DETAILS OF MODEL STRUCTURE AND DISCRIMINATORS

**Encoder & Decoder Architecture**  The encoder is constructed as a sequential series of components: starting with a 1D convolutional layer featuring $C$ channels and a kernel size of 7, followed by a set of $B$ residual conventional blocks. Each block is composed of two dilated convolutions with dilation rate of $(1, 1)$ and kernel size of $(3, 1)$ and a skip-connection, followed by a strided convolutional down-sampling layer, with a kernel size $K$ of the twice the stride $R$. Whenever down-sampling, the number of channels is doubled. Unlike in EnCodec that the convolution blocks are followed by a two-layer LSTM, we use BiLSTM to augment the semantic modeling ability. A final 1D convolution layer with a kernel size of 7 is used to set the dimensionality of embeddings to $D$. We use $C = 32, B = 4$ and $(2, 4, 5, 8)$ as strides. We use ELU (Clevert et al., 2016) as a non-linear activation either layer normalization (Ba et al., 2016) or weight normalization (Salimans & Kingma, 2016).The decoder mirrors the encoder and uses transposed convolutions and LSTM instead of stride convolutions and BiLSTM, with the strides in reverse order as in the encoder. The decoder outputs the final audio signal.

**Residual Vector Quantizer**  We use Residual Vector Quantizer (RVQ) to quantize the encoder output and follow the same training procedure as EnCodec. During training, the code selected for each input is updated using an exponential moving average with a decay of 0.99, and codes which have not been assigned any input vector for several batches are replaced with input vectors randomly sampled within current batch. Straight-through-estimator (Bengio et al., 2013) is used to compute the gradient of encoder, e.g. as if the quantization step was the identity function during the backward phase. Finally, a commitment loss, consisting of the MSE between the input of the quantizer and its output, with gradient only computed with respect to its input, is added to the overall training loss.

**Discriminator**  The MS-STFT discriminator utilizes networks with identical structures that operate on multi-scaled complex-valued STFT, where the real and imaginary parts are concatenated. For each sub-network, it is composed of a 2D convolutional layer (using kernel size $3 \times 8$ with 32 channels), followed by 2D convolutions with increasing dilation rates in the time dimension (1, 2 and 4), and a stride of 2 over the frequency axis. A final 2D convolution with kernel size $3 \times 3$ and stride $(1, 1)$ provide the final prediction. For MSD and MPD, we follow the same settings as in HiFiGAN (Kong et al., 2020) but adjust the channel number to align the discriminator's parameters more closely with that of MS-STFT.

## E    RELATED WORK

**Discrete Speech Representations**  There are two popular speech discrete representations: semantic tokens and acoustic tokens. Semantic tokens can be extracted from self-supervised learning of speech representations (Hsu et al., 2021; Chung et al., 2021) and encode high-level representations that correlate with coarse, symbolic features while paralinguistic information such as speaker identity and acoustic details are removed. Acoustic tokens can be extracted from neural audio codec (Zeghidour et al., 2021; Défossez et al., 2022; Yang et al., 2023) and provide high-fidelity reconstruction of the acoustic details. But they can not decouple different information of speech. SpeechTokenizer unifies the two types of tokens, enabling both high-quality audio reconstruction and decomposition of different information of speech.

**Spoken Generative Language Models**  Speech discrete representation based spoken generative language models have demonstrated remarkable performance on various speech processing tasks (Borsos et al., 2022; Wang et al., 2023; Kharitonov et al., 2023; Zhang et al., 2023). AudioLM (Borsos et al., 2022) proposes to model speech based on audio codecs together with semantic codes, which can synthesize speech in a textlesss setting. VALL-E (Wang et al., 2023) leverages neural codec models to represent speech in discrete tokens from eight quantizers. VALL-E comprises of an autoregressive language model that converts phoenmes to acoustic tokens from the first quantizer and an non-autoregressive language model to generate codes of the other seven quantizers. However, VALL-E suffers from problems that some words may be unclear, missed, or duplicated in speech synthesis due to the information gap between acoustic tokens and phoneme. To bridge the gap, SPEAR-TTS (Kharitonov et al., 2023) uses semantic tokens as a bridge between text and acoustic tokens. It first generates semantic tokens from text and then produces acoustic tokens from semantic tokens. However, this multi-stage modeling approach is more complex and can lead to problems like error accumulation and slow inference speed. The first quantizer of SpeechTokenizer generates semantic tokens, while the remaining seven quantizers produce acoustic tokens by modeling the paralinguistic information lost in the semantic tokens. SpeechTokenizer-based VALL-E combines the advantages of VALL-E and SPEAR-TTS, where the autoregressive model can perform text-to-semantic tokens conversion, and the non-autoregressive model can achieve semantic-to-acoustic tokens conversion.

**Speech Representation Disentanglement**  Human speech can be roughly decomposed into three components: content, timbre, and prosody (Liu et al., 2023). Content represents the main information in the speech, which can be expressed using text or phonemes. Timbre represents the speaker's characteristics, while prosody encompasses intonation, stress, and rhythm of speech, reflecting how the speaker conveys the content information. Current Speech Representation Disentanglement (SRD) methods mostly separate speaker information from content information for voice conversion (Qian et al., 2019; Casanova et al., 2022). These approaches adopt a parallel disentanglement strategy, where the speech is fed into parallel content and speaker encoders to obtain different representations (Qian et al., 2020). However, this strategy heavily relies on prior knowledge and introduces strong inductive biases, making the modeling process more complex and potentially overlooking certain speech information like prosody. Differently, VQVC (Wu & Lee, 2020) models the content embedding as a series of discrete codes and take the difference between quantize-before and quantize-after vector as the speaker embedding. Similarly, SpeechTokenizer utilizes a residual structure to perform serial decomposition of speech information and models different information as discrete tokens.

## F    CODEBOOK ANALYSIS

We investigate whether the tokens learned by the first RVQ quantizer relate to phonetic information. Utilizing SpeechTokenizer or EnCodec, we derive speech tokens from the TIMIT training set and extract the RVQ-1 tokens, denoted as $q_1$. We then compute the conditional probability $p(phoneme|q_1)$ based on the co-occurrence between phonemes and the codes. The alignments are constructed by selecting the phoneme that occurs most frequently in the receptive field for each $q_1$.

Figure 4 visualizes the conditional probability $p(\text{phoneme}|q_1)$ for both SpeechTokenizer and EnCodec. A darker color block indicates a higher $p(\text{phoneme}|q_1)$. A more distinct contrast between the diagonal color band and its surrounding area signifies greater phoneme purity, which in turn suggests a more accurate mapping between the code and its corresponding phoneme. For SpeechTokenizer, it's evident that in the codebook of RVQ-1 quantizer, many discrete codes seem to specialize in capturing specific phonetic sounds, indicating RVQ-1 quantizer can obtain a good alignment between codes and labeled phonemes. However, for EnCodec, this phenomenon is not as obvious.

Additionally, Figure 4 also reveals that over 600 codes from the EnCodec RVQ-1 codebook have never been utilized, suggesting a suboptimal utilization rate of the codebook when EnCodec encodes speech. A lower utilization rate of the codebook implies that more RVQ layers are required to ensure the quality of synthesized speech, consequently necessitating the generation of more codes during the construction of a spoken generative language model, resulting in greater space, time and computation power consumption.

We further evaluate the models using Phone-Normalized Mutual Information (PNMI) (Hsu et al., 2021). As shown in Table 8, RVQ-1 tokens of SpeechTokenizer achieve a superior PNMI score to that of HuBERT units and significantly outperforms EnCodec-RVQ-1. This suggests that the semantic distillation process in SpeechTokenizer is effective, thereby explaining its enhanced text alignment performance.

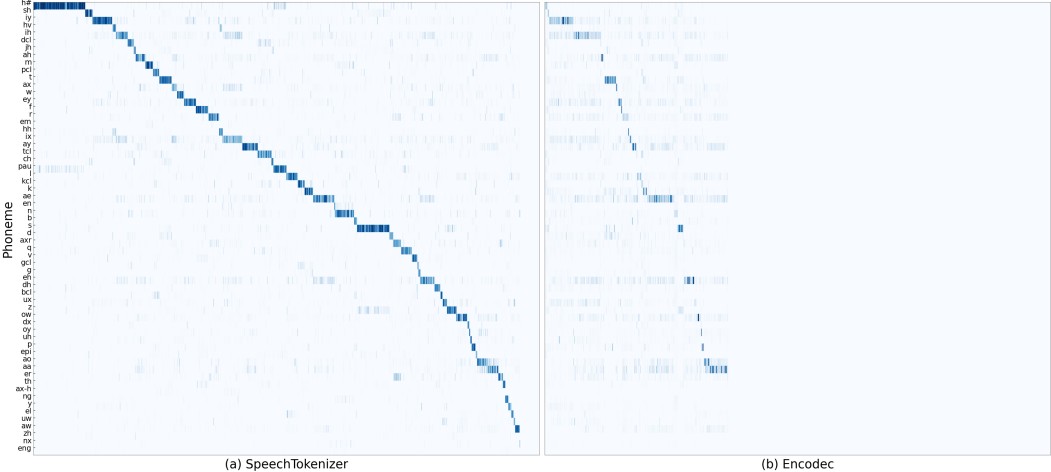

Figure 4: Visualization of the conditional probability $P(\text{phoneme}|\text{code})$ on TIMIT train set. The y-axis is the phoneme set and the x-axis is the codewords of the first RVQ layer sorted by the most correlated phoneme.

| Tokenizer | | PNMI↑ |
|---|---|---|
| HuBERT | KM500 | 0.43 |
| EnCodec | RVQ-1 | 0.28 |
| SpeechTokenizer | RVQ-1 | 0.71 |

Table 8: PNMI of different discrete speech representation.

## G    EXTENSION TO UNSEEN LANGUAGE

Since the paralinguistic information is considered to be language-agnostic, we attempted to apply SpeechTokenizer directly to unseen languages. We choose German and Chinese. For German, we select samples from the German subset of Multilingual LibriSpeech dataset for testing. For Chinese, we select samples from the Aishell-3 dataset (Shi et al., 2021) for testing. We resynthesize speech from RVQ-1 and RVQ-1:8 tokens. Resynthesized speech are displayed in our demo website [1]. We also analysis the melspectrogram of German speech and English speech in Appendix H.

Results show that for languages either closely or distantly related to English, resynthesized speech from RVQ-1 tokens tend to lose timbre and prosody information while maintaining clear content. The resynthesized speech generated from RVQ-1:8 tokens is very close to the grountruth. That suggests SpeechTokenizer can achieve hierarchical information disentanglement on unseen language, even though SpeechTokenizer is trained solely on English data. We believe that SpeechTokenizer may possess the ability to extract content from speech while disregarding language-dependent features. This ability holds promise for the development of a multilingual SpeechTokenizer.

---

[1]`https://0nutation.github.io/SpeechTokenizer.github.io/`

# H   MELSPECTORGRAM ANALYSIS

We plot the melspectrogram of raw speech, resynthesized speech of EnCodec RVQ-1 tokens, and resynthesized speech of SpeechTokenizer RVQ-1 tokens. From the figure 5, it's evident that the melspectrogram corresponding to EnCodec RVQ-1 largely retains the stripes and shapes in the raw melspectrogram. In contrast, the speech resynthesized from SpeechTokenizer RVQ-1 essentially loses all of the horizontal stripes, which indicates that timbre and prosody information has been diminished.

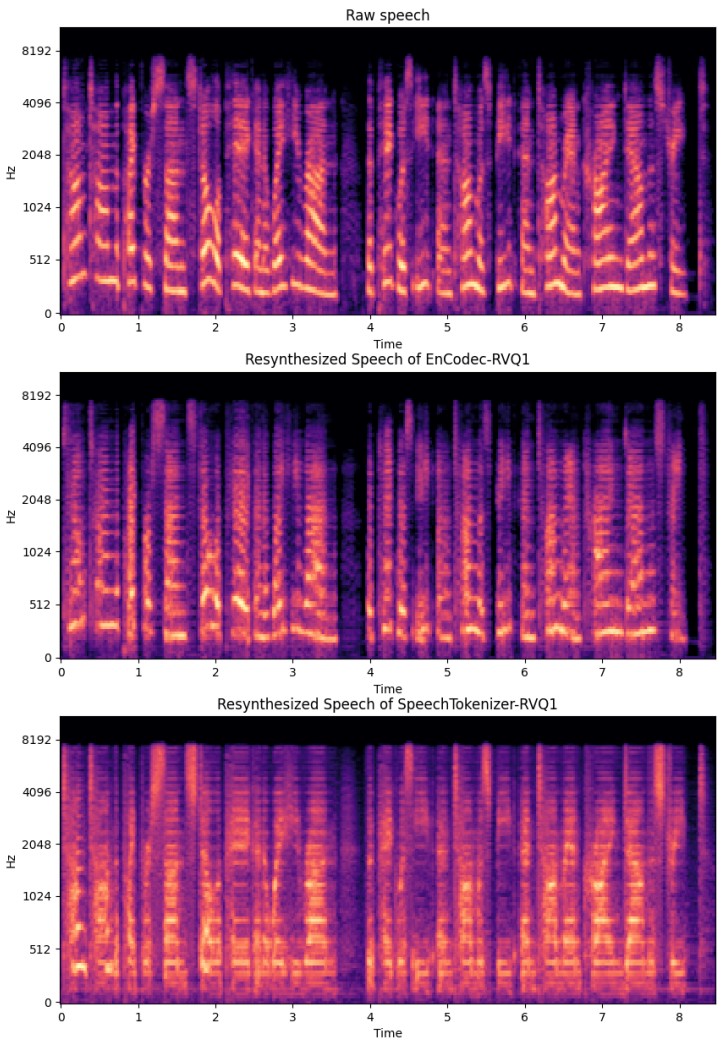

Figure 5: Melspectorgram of raw speech, resynthesized speech of SpeechTokenizer and EnCodec RVQ-1 tokens.

We alse plot melspectrogram of raw German speech and resynthesized German speech of SpeechTokenizer RVQ-1 tokens. As shown in the Figure 6, the same patterns observed in English speech are also present in German speech.

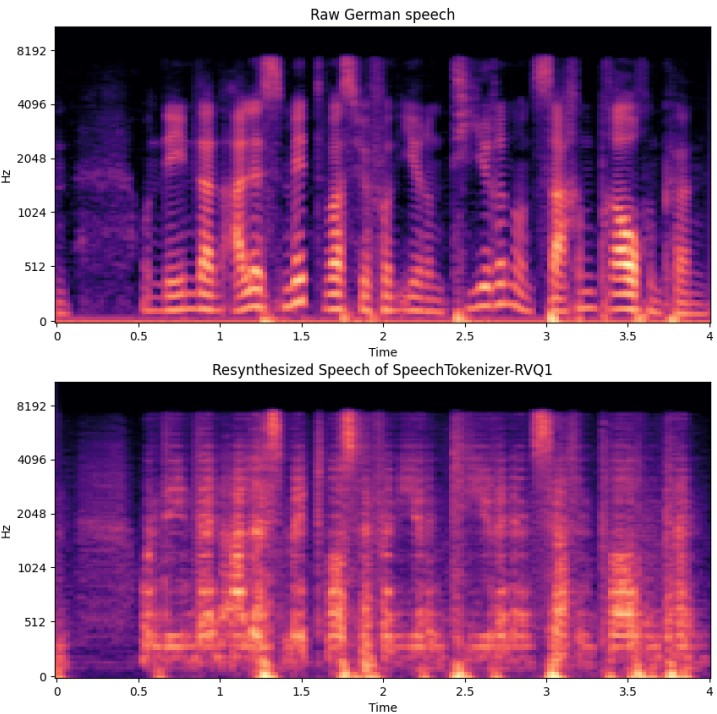

Figure 6: Melspectorgram of German speech and resynthesized speech of SpeechTokenizer RVQ-1 tokens.

