# OpenReview forum: "SpeechTokenizer: Unified Speech Tokenizer for Speech Language Models"
_ICLR.cc/2024/Conference — ICLR 2024 poster_

### Official Review · Reviewer_g1yJ · 2023-10-26

**Soundness:** 2 fair
**Presentation:** 3 good
**Contribution:** 3 good
**Rating:** 6
**Confidence:** 4

**Summary:**

The paper mainly makes two parts of contributions. The first one is a benchmark, SLMTokBench, which assesses the suitability of speech tokens for building speech language models. The second one, which is also the main contribution of the paper, is a unified speech tokenizer that combines both semantic and acoustic tokens. SpeechTokenizer performs comparably with pure acoustic token (EnCodec) in speech reconstruction, and its performance is much higher than SpeechTokenizer in VALL-E.

**Strengths:**

1. The paper proposes a novel method to unify the semantic tokens and acoustic tokens.

2. The experiments show that the unified tokenizer, SpeechTokenizer, large improves the acoustic tokens from EnCodec in the zero-shot TTS experiment in VALL-E. The improvements are significant in both objective metrics and subjective metrics.

3. The proposed benchmark, SLMTokBench, is well motivated and reasonable.

4. The paper is well written and easy to follow.

**Weaknesses:**

1. My first concern is about the necessity for a unified speech tokenizer. In the last several sentence of page 1, the authors argue that "the multi-stage modeling approach is more complex, leading to several drawbacks such as error accumulation and slower processing speed". Is there any evidence to support the claim? In addition, I may somewhat disagree with how the paper describe the characteristic of "Semantic LM and Acoustic LM" in Table 1. In my view, Semantic LM does not generate Accurate Content but it is speed is much faster. And Acoustic LM does generate Accurate Content.

2. The second concern, which is also my major concern, is about whether the complicated training pipeline really leads to better performance in zero-shot TTS than Hierarchical LM. If using VALL-E with the semantic token for HuBERT and RVQ1:7 from EnCodec (It has the same bit rate as USLM), what is performance?

3.  At last, I am doubtful whether the SpeechTokenizer only works for AR + NAR model like VALL-E. As VALL-E only uses the first speech token in the AR model, putting more information on the first token may be beneficial to its performance so that the improvement of USLM in zero-shot TTS is much larger than the improvements of SpeechTokenzier in speech reconstruction. I am not sure such large improvements will still exist in Hierarchical model like AudioLM.

I understand that weakness 3 is hard to verify during the rebuttal. Therefore, If the authors can resolve my first two concern, I am willing to raise my score.

**Questions:**

Can the author provide some explainations why the improvement of USLM in zero-shot TTS is much larger than the improvements of SpeechTokenzier in speech reconstruction (Table 4 v.s. Table 2) ?

**Details Of Ethics Concerns:**

The technique of the paper may be used to produce fake speech of humans, which I believe could potentially harm the society.

---

> ### Author Response · Authors · 2023-11-22
>
> Thanks for your careful and valuable comments. We will explain your concerns point by point.
> ### Q1: Necessity for a unified speech tokenizer
> AudioLM [1] is a representative of Hierarchical LMs which is composed of three cascaded autoregressive models: Semantic modeling, Coarse acoustic modeling, and Fine acoustic modeling. The complex tri-stage autoregressive cascade approach is more prone to error accumulation and introduces a high time complexity. Recently, SoundStorm [2] replaces the last two stages of AudioLM with one NAR model, and the experimental results show they alleviating the issues of slow processing speed, and achieves better audio generation quality. However, the redundancy of information between semantic tokens and acoustic tokens adds unnecessary complexity and difficulty to the NAR modeling. So we proposed SpeechTokenizer, by sequentially decoupling semantic and acoustic information within the RVQ1 and RVQ2:8 frameworks, hoping to reduce the modeling difficulty of NAR. This has been proven in our Hierarchical LM comparative experiments.
>
>  In terms of the advantage of Semantic LM over Acoustic LM in generating accurate content, the AudioLM [1] paper states, 'Using the model trained only on the acoustic tokens, we sample speech continuations from a prompt of 4 seconds. While both the recording conditions and the speaker identity from the prompt are preserved, the linguistic content is inconsistent, and often akin to babbling.' This demonstrates the lack of superiority of Acoustic LM in generating accurate content. Furthermore, our paper's Section 2.3 analysis reveals that in SLMTokenBench, semantic tokens have better text alignment than acoustic tokens, suggesting a more effective mapping relationship with the content for semantic tokens.
> ### Q2: Comparsion with Hierarchical LM
> We implemented a Hierarchical LM that combines Semantic token modeling with SoundStorm, and compared its performance with USLM's in zero-shot TTS, under the same data and experimental settings. We use HuBERT L9 units as semantic tokens and codes generated by EnCodec as acoustic tokens here. Given that the open-source EnCodec and HuBERT have different sampling rate, and aslo in order to ensure a fairer comparison, we used our own version of EnCodec trained on LibriSpeech, maintaining consistency with SpeechTokenizer. At the same time, we adjusted USLM to the same architecture, where we used SpeechTokenizer RVQ-1 for Semantic modeling, and SoundStorm then generates the remaining RVQ2-8 tokens conditioned on the input RVQ1 tokens. We followed the original paper in implementing SoundStorm’s model structure, mask scheme, and decoding strategy. We conduct evaluation on the VCTK dataset and the results are summarized in the Table below.
> |  | AR (Transformer Decoder) |  | NAR（SoundStorm） |  | Evaluation |  |
> | --- | --- | --- | --- | --- | --- | --- |
> | Exp | Input | Output | Input | Output | WER↓  | Speaker Similarity ↑ |
> | 1 USLM | text | SpeechTokenizer RVQ-1 | SpeechTokenizer RVQ-1 | SpeechTokenizer RVQ-2:8 | 4.49 | 0.88 |
> | 2 Hierarchical LM | text | HuBERT units | HuBERT units | Encodec RVQ-1:8 | 7.18 | 0.79 |
> | 3 Hierarchical LM | text | HuBERT units | HuBERT units | Encodec RVQ-1:7 | 7.5 | 0.56 |
> | 4 Hierarchical LM | text | HuBERT units | HuBERT units | SpeechTokenizer RVQ-1:8 | 5.94 | 0.81 |
>
> As shown in the Table, USLM (Exp 1) significantly outperforms Hierarchical LM (Exp 2 3 4) in both WER and Speaker Similarity. We attribute this improvement to the decoupling of information by SpeechTokenizer. In Hierarchical LM, the modeling process from HuBERT units to acoustic tokens still requires modeling all the information of speech in the acoustic tokens, which increases the modeling difficulty. However, USLM’s NAR model receives RVQ-1 tokens as input and needs to output RVQ-2:8 tokens that contain only paralinguistic information, significantly reducing the modeling complexity.
> We also observed that in Hierarchical LMs, generating only EnCodec's RVQ1-7 (Exp 3) leads to a decline in model performance, with an increase in WER and a decrease in Speaker Similarity.
>
> [1] Borsos, Zalán, et al. "Audiolm: a language modeling approach to audio generation." IEEE/ACM Transactions on Audio, Speech, and Language Processing (2023).
>
> [2] Borsos, Zalán, et al. "SoundStorm: Efficient Parallel Audio Generation." arXiv preprint arXiv:2305.09636 (2023).

---

> ### Author Response · Authors · 2023-11-22
>
> ### Q3: Applying SpeechTokenizer to Hierarchical LM
> As shown in the Table above, Hierarchical LM with SpeechTokenizer (Exp 4) yields better results compared to with Encodec (Exp 2). We think this is partly because the task of HuBERT unit -> Speechtokenizer RVQ1 is simpler than HuBERT unit -> Encodec. Moreover, due to NAR's layered modeling of RVQ, the process of NAR modeling different layers of RVQ in Speechtokenizer is implicitly decoupled into two relatively independent processes: semantic modeling (HuBERT units -> SpeechTokenizer RVQ-1) and paralinguistic modeling (HuBERT units + SpeechTokenizer RVQ-1:(i - 1) -> RVQ-i). This reduces the complexity of modeling, enabling Hierarchical LM to achieve better results with Speechtokenizer than with Encodec.
> ### Q4: Explanation of larger improvement of USLM in zero-shot TTS
> We attribute the performance improvement of USLM in zero-shot TTS to the information decoupling feature in SpeechTokenizer, rather than just improvements in speech reconstruction quality. Concretely, the information disentanglement strategy in SpeechTokenizer significantly simplifies both autoregressive (AR) and non-autoregressive (NAR) modeling in zero-shot TTS.
>
> Firstly, for AR models, the RVQ-1 tokens in SpeechTokenizer align more easily with text compared to RVQ-1 tokens in EnCodec. In the codebook analysis section of our paper's Appendix F, we visualized the alignment between the phones and codes of SpeechTokenizer's and EnCodec's RVQ-1, and computed pnmi. It is clearly evident that the alignment of SpeechTokenizer's codes with phones is significantly higher than that of EnCodec. Therefore, theoretically, it is simpler for the model to learn phoneme to SpeechTokenizer RVQ-1 mapping than phoneme to EnCodec RVQ-1.
>
> Secondly, for non-autoregressive (NAR) models, the RVQ-2:8 tokens in SpeechTokenizer only contain acoustic details of the speech. This contrasts with EnCodec's RVQ-2:8 tokens, where various types of information are intermingled, making the tokens more difficult to predict. So the NAR model in USLM performs an easier task.
>
> During the AR and NARtraining process, we recorded the model's top10 accuracy in predicting tokens on the validation set, as shown in the following table. USLM achieved a higher top10 accuracy than VALL-E in both AR and NAR models. This result validates our intuition.
> |  | AR | NAR |
> | --- | --- | --- |
> | VALL-E | 0.77 | 0.57 |
> | USLM | 0.99 | 0.7 |

---

> ### Comment · Reviewer_g1yJ · 2023-11-23
> **Follow-up questions**
>
> Thanks for the detailed response from the authors. I just have additional questions about the experiments. Why is the WER of USLM in Q2 is different from the reported number in Table 4. Additionally, why is the NAR part of the model is SoundStream, is it different from NAR of VALL-E in the main paper?

---

> ### Author Response · Authors · 2023-11-23
>
> Thank you for your insightful comments and questions. We appreciate the opportunity to clarify and elaborate on these points to address your concerns more effectively.
> ### Q5: Why is the WER of USLM in Q2 different from the reported number in Table 4?
> In Q2, the model structure of USLM differs from that in Table 4. In Table 4, the architecture of USLM is the same as VALL-E, while in Q2, it is consistent with Hierarchical LM. The NAR part in the former is a transformer encoder, while in the latter, it's a Conformer.
> ### Q6: Why the NAR part of the model is SoundStorm, and how does it differ from VALL-E's NAR in the main paper?
> We used SoundStorm in Hierarchical LM for modeling from semantic tokens to acoustic tokens because the experiments in SoundStorm demonstrated its superiority over the two-stage autoregressive (AR) modeling in AudioLM in terms of speed and performance. The USLM we refer to is a concept rather than a specific structure, as illustrated in Figure 1(b), utilizing SpeechTokenizer's disentangling feature for layered information modeling. This involves using AR on RVQ-1 for semantic information and a NAR-like model on RVQ-2:8 for prosodic information beyond semantics. In the paper, for a fair comparison between SpeechTokenizer and EnCodec as representatives of acoustic tokens in building speech language models, we set USLM to have the same model structure and experimental setup as VALL-E. In the Q2 experiment, for a fair comparison between SpeechTokenizer and a combination of semantic tokens + acoustic tokens in building speech language models, we used the same model structure and experimental setup as Hierarchical LM. In both experimental setups, USLM outperformed its counterparts, demonstrating that SpeechTokenizer is more suitable for building speech language models than both acoustic tokens and a combination of semantic tokens + acoustic tokens.
>
> SoundStorm and VALL-E's NAR are different. VALL-E’s NAR model structure is a transformer encoder, whereas SoundStorm’s structure is Conformer.

---

> ### Comment · Reviewer_g1yJ · 2023-11-23
> **Follow-up questions**
>
> Thanks for follow-up response from the authors. While I feel the experiments in the rebuttal is detailed and resolve most of my concern, using Soundstorm instead of VALL-E is really strange for comparing Hierarchical LM and USLM is strange. By taking into other reviewers' comments, I would increase my rating to 6.

---

> > ### Author Response · Authors · 2023-11-23
> >
> > Thank you very much for your valuable feedback. Your insightful questions and suggestions have greatly aided in improving our work!
> >
> > The choice of SoundStorm in Hierarchical LM was driven by existing literature and experiments that have validated its superior performance in this domain. On the other hand, the performance of VALL-E's architecture within Hierarchical LM has not been sufficiently verified. We hence chose to use SoundStorm in Hierarchical LM so as not to be considered as having chosen a weaker baseline for comparison.
> >
> > Thank you once again for your valuable comments and suggestions!

---

### Official Review · Reviewer_usJL · 2023-10-28

**Soundness:** 2 fair
**Presentation:** 2 fair
**Contribution:** 2 fair
**Rating:** 6
**Confidence:** 3

**Summary:**

This paper proposed SpeechTokenizer to learn unified speech tokenizer for language models.  Experiments demonstrate that the Unified Speech Language Model (USLM) leveraging SpeechTokenizer SpeechTokenizer shows comparable performance when compared with EnCodec in speech reconstruction and SLMTokBench benchmark. In zero-shot Text-to-Speech tasks, USLM achieves better performance than VALL-E.

**Strengths:**

1. The motivation of the paper is straightforward and push the boundary of speech language models to more unified representation for different tasks.
2. Extensive experiments are done to validate the performance comparison.

**Weaknesses:**

1. The paper misses the comparison with Hierarchical speech language models as listed in the paper. if the comparison could be added, it would be more convincing.
2. Although the motivation of the paper is a shining point for the paper, the technical contribution of the paper is limited as the framework is borrowed from existing work, e..g, the framework of RVQ-GANs. If authors could propose some modifications on the training strategy or the model architectures based on some observations from experiments, it would add more insightful points.

If my above concerns are resolved, I would consider increasing my rating.

**Questions:**

My questions are listed above.

---

> ### Author Response · Authors · 2023-11-22
>
> Thanks for your careful and valuable comments. We will explain your concerns point by point.
> ### Q1: Comparision with Hierarchical speech language model
> We implemented a Hierarchical LM that combines Semantic token modeling with SoundStorm [1], and compared its performance with USLM's in zero-shot TTS, using the same data and experimental settings. We use HuBERT L9 units as semantic tokens and codes generated by EnCodec as acoustic tokens here. Given that the open-source EnCodec and HuBERT have different sampling rate, and aslo in order to ensure a fair comparison, we used our own version of EnCodec trained on LibriSpeech, maintaining consistency with SpeechTokenizer. At the same time, we adjusted USLM to the same architecture, where we used SpeechTokenizer RVQ-1 for Semantic modeling, and SoundStorm then generates the remaining RVQ2-8 tokens conditioned on the input RVQ1 tokens. We followed the original paper in implementing SoundStorm’s model structure, mask scheme, and decoding strategy. We conduct evaluation on the VCTK dataset and the results are summarized in the Table below.
> |  | AR (Transformer Decoder) |  | NAR（SoundStorm） |  | Evaluation |  |
> | --- | --- | --- | --- | --- | --- | --- |
> | Exp | Input | Output | Input | Output | WER↓  | Speaker Similarity↑ |
> | USLM | text | SpeechTokenizer RVQ-1 | SpeechTokenizer RVQ-1 | SpeechTokenizer RVQ-2:8 | 4.49 | 0.88 |
> | Hierarchical LM | text | HuBERT units | HuBERT units | Encodec RVQ-1:8 | 7.18 | 0.79 |
>
> As shown in the Table, USLM significantly outperforms Hierarchical LM in both WER and Speaker Similarity. We attribute this improvement to the decoupling of information by SpeechTokenizer. In Hierarchical LM, the modeling process from HuBERT units to acoustic tokens still requires modeling all the information of speech in the acoustic tokens, which increases the modeling difficulty. However, USLM’s NAR model receives RVQ-1 tokens as input and needs to output RVQ-2:8 tokens that contain only paralinguistic information, significantly reducing the modeling complexity.
>
> ### Q2: More technical contribution
> 1. We introduced a novel sequence distillation loss, and its effectiveness was validated through ablation experiments in Appendix C. The results demonstrate that this approach significantly enhances the sequence modeling capabilities of RVQ-1 tokens. Our proposed sequence distillation loss holds the potential to significantly enhance performance in a variety of sequence distillation tasks.
> 2. We proposed a new paradigm for speech representation disentanglement—serial disentanglement. Current approaches primarily use a parallel disentanglement strategy, simultaneously processing speech through separate content, speaker encoders, etc., for extracting different representations and achieving disentanglement. However, this method heavily relies on prior knowledge and introduces strong inductive biases, increasing complexity and potentially overlooking certain speech aspects like prosody. We adopt residual structure to achieve serial disentanglement of speech, offering a novel perspective for  information disentangled representation learning.
> 3. In our experiments, we observed that RVQ-1 requires contextual modeling capabilities. To address this, we replaced LSTM in EnCodec with BiLSTM, and the effectiveness of BiLSTM was validated through ablation experiments in Appendix B.
>
> [1] Borsos, Zalán, et al. "SoundStorm: Efficient Parallel Audio Generation." arXiv preprint arXiv:2305.09636 (2023).

---

### Official Review · Reviewer_pbGo · 2023-10-29

**Soundness:** 3 good
**Presentation:** 3 good
**Contribution:** 2 fair
**Rating:** 3
**Confidence:** 4

**Summary:**

The paper addresses the limitations of current speech representations in large language models. It introduces a benchmark called SLMTokBench to assess the suitability of existing speech tokens for speech language modeling. The results indicate that neither semantic nor acoustic tokens are ideal for this purpose. To overcome this, the authors propose SpeechTokenizer, a unified speech tokenizer for speech large language models.

SpeechTokenizer utilizes the Encoder-Decoder architecture with residual vector quantization (RVQ) and combines semantic and acoustic tokens. It disentangles different aspects of speech information hierarchically across RVQ layers, providing a more comprehensive representation of speech data. The authors also construct a Unified Speech Language Model (USLM) using SpeechTokenizer.

Experimental results demonstrate that SpeechTokenizer performs comparably to EnCodec in speech reconstruction and shows strong performance on the SLMTokBench benchmark. Overall, the paper introduces SpeechTokenizer as a solution for improving speech language models by addressing the limitations of existing speech tokens.

**Strengths:**

1. The motivation of the paper is strong. Speech research area needs speech specific fundamental innovation. Many papers simply borrow ideas from NLP and CV and test their performance on speech data. Exploring how to discretize continuous speech signals is an intriguing concept. The authors propose a simple and logical approach, which serves as a promising initial step for the field.


2. SLMTOKBENCH offers valuable insights for evaluating current speech tokenization methods. It has the potential to become a fundamental benchmark for speech tokenization research. However, it still requires some improvements, as mentioned in the weaknesses . section.

**Weaknesses:**

1. My primary concern lies with the experimental section of the paper. Firstly, the authors trained SpeechTokenizer on the LibriSpeech set, while the EnCodec was trained on a mixed dataset of 10,000 hours. As a result, EnCodec should possess better generalization capabilities for speech reconstruction, particularly in the presence of noise. However, the authors only evaluated the performance on LibriSpeech, which is an in-domain test set for their method but falls outside the domain of EnCodec. This unfair experiment fails to demonstrate whether their method outperforms EnCodec. A fair experiment should involve testing the model on various test sets, especially those unseen during training.

SLMTOKBENCH should encompass a range of speech environments, not limited to LibriSpeech alone. Audiobooks represent just one type of speech data, and we should consider incorporating a more diverse test set for a comprehensive evaluation.


2.How did you arrive at the WER number of 7.9 in Table 4? I recall that the VALL-E paper reported a WER of around 3.X on LibriSpeech.
Additionally, other studies on LibriSpeech have achieved comparable or even lower WER values.

3. In the context of the speech tokenizer, it is necessary to take into account additional baselines like SoundStream and the multi-band diffusion-based method, rather than solely comparing it with Encode.

**Questions:**

Why the authors not directly cite the number in VALL-E paper for evaluation? I don't think it is a good idea to re-train VALL-E on a smaller training set.

---

> ### Author Response · Authors · 2023-11-22
>
> Thank you for your valuable suggestions on our experimental setting. We would like to clarify that the purpose of proposing SpeechTokenizer is not to achieve a Codec model that surpasses EnCodec in terms of speech reconstruction quality or robustness. Instead, our goal is to provide a more suitable speech discretization tool for building speech language models by unifying semantic tokens and acoustic tokens. Therefore, we train it on LibriSpeech but not a large and diverse speech dataset to develop a high-performance codec. Our experimental evaluation focuses on the disentanglement effects between different RVQ layers of SpeechTokenizer and its performance in speech language modeling. Due to space limitations, we did not design extensive and comprehensive evaluation experiments for codec-related tasks.
> ### Q1: comparision with Encodec:
> We evaluate SpeechTokenizer on the VCTK dataset, which is beyond our training set. As indicated in the table below, the experimental results show that SpeechTokenizer outperforms EnCodec on VCTK. Moreover, we noticed a recent paper [1] mentioning that incorporating semantic information into EnCodec enhances sound quality. This finding aligns with our experimental observations, further validating the effectiveness of our approach.
> |  | VCTK |  |
> | --- | --- | --- |
> |  | WER↓  | VISQOL↑ |
> | EnCodec(official) | 3.6 | 4.14 |
> | SpeechTokenizer | 3.7 | 4.30 |
>
> ### Q2: Explanation of WER number of 7.9 of VALL-E:
> As VALL-E's model is not open-sourced, to ensure a fair comparison with USLM, we retrained VALL-E under the same experimental setup and data conditions. We believe there are two main reasons for the discrepancy between our self-trained version and the WER reported in the original VALL-E paper:
> - Differences in the test set and prompt selection method: The original VALL-E [2] paper distinguished between VALL-E-continual and VALL-E based on whether the prompts and texts are semantically continuous. VALL-E-continual achieved a WER of 3.8 on the LibriSpeech dataset, compared to 5.9 for VALL-E. In our experiments, we did not use the continual prompt method, thus the appropriate WER to compare with is 5.9, not 3.8. Additionally, our test set was the VCTK dataset, for which VALL-E's paper did not report zero-shot TTS WER. However, they noted in the limitations section that 'The worse result on VCTK than LibriSpeech also implies insufficient coverage of accent speakers.' This suggests that VALL-E's performance on VCTK might be inferior to that on LibriSpeech. Therefore, it is reasonable that our self-trained VALL-E achieved a WER of 7.9 on VCTK, which is slightly weaker than its 5.9 performance on LibriSpeech.
> - Differences in training data: The original VALL-E paper used a 60,000-hour LibriLight dataset for training, while we used a 40,000-hour multilingual LibriSpeech dataset. This difference in the amount of training data could lead to variations in performance.
> ### Q3: Additional baselines like SoundStream
> Since SoundStream and the multi-band diffusion-based codec do not have official open-source versions available, we are unable to directly compare our approach with these baselines.
> ### Q4: not directly citing the number in VALL-E paper for evaluation
> Since the VALL-E paper only report WER results on the LibriSpeech dataset, to cite numbers from the VALL-E paper and ensure a fair comparison, we evaluate the zero-shot TTS performance of USLM on the LibriSpeech dataset. Our evaluation experiment setting was completely identical to that in the VALL-E paper, using non-continual 3-second audio clips as prompts. The results are shown in the following table.
> |  | WER↓  | SIM↑  |
> | --- | --- | --- |
> | VALL-E report | 5.9 | 0.580 |
> | USLM | 5.4 | 0.675 |
>
>  As can be seen, USLM outperforms VALL-E in both WER and SIM. Furthermore, we have provided audio demos on our demo page, where the superiority of our approach over VALL-E can be clearly heard.
>
> [1] Du, Zhihao, et al. "FunCodec: A Fundamental, Reproducible and Integrable Open-source Toolkit for Neural Speech Codec." arXiv preprint arXiv:2309.07405 (2023).
>
> [2] Wang, Chengyi, et al. "Neural codec language models are zero-shot text to speech synthesizers." arXiv preprint arXiv:2301.02111 (2023).

---

### Official Review · Reviewer_tD35 · 2023-11-01

**Soundness:** 3 good
**Presentation:** 3 good
**Contribution:** 3 good
**Rating:** 8
**Confidence:** 4

**Summary:**

This paper propose a new model to learn a speech tokenizer that can both reconstruct speech well and retain enough semantic information in speech. It simplifies previous work that need to use separate tokenizers for semantic and acoustic information. Since good tokenizers play an important role in speech language models, the results in this paper could benefit the advancement of research in speech language models in the LLM era.

**Strengths:**

Originality: this is the first work trying to inject semantic information to acoustic tokens based on soundstream/encodec which has been successfully applied to speech language modeling, TTS, etc. Though the idea is kind of straight, this paper still deserves originality.
Quality: Overall the solution is clearly motivated and reasonably implemented, and corresponding evaluations are comprehensive.
Clarity: the paper is easy to read, the results are easy to understand.
Significance: A good tokenizer is important for many downstream tasks, especially for speech generation tasks. This study shows it's possible to use a single model as tokenizer and achieve comparable or even better performance than previous studies using combination of different tokenizers encoding different aspects of speech

**Weaknesses:**

1. It's not quite clearly explained why the authors chose the first RVQ to inject semantic information. Why not the later ones or even for all RVQs
2. It could be better if the authors can show some experiments using the new tokenizers for speech understanding tasks like recognition/speech translation with relatively large scale training data, and compare to Hubert/soundstream/encodec tokens.

**Questions:**

1. Is it possible to directly combine the training loss of hubert and soundstream/encodec instead of distillation to achieve unified speech tokenizer?
2. Since soundstream/encodec is a small model, is it possible the model can learn good representations when the model is large enough?

---

> ### Author Response · Authors · 2023-11-22
>
> Thank you for recognizing the value of our work and for your insightful comments. Below are our responses to your questions.
> ### Q1: Explanation of  choosing the first RVQ to inject semantic information
> We chose to inject semantic information into the first RVQ because our goal was to unify the semantic token (hubert token) and the acoustic token (encoded token) within a single framework. Since the HuBERT token is a one-dimensional sequence, we decided to inject semantic information into only one RVQ layer to align it with the semantic token. Additionally, restricting semantic information to only RVQ-1 means that in the USLM's AR model, only the RVQ-1 token needs to be predicted, avoiding the issue of predicting long sequences.
> ### Q2: For speech understanding tasks
> In our subsequent work, we plan to utilize the tokenizer with a substantial amount of data to develop a model capable of both speech understanding and speech generation.
> ### Q3: Use training loss of HuBERT
> We believe that directly combining the masked language modeling loss used in pre-trained Hubert with the reconstruction loss of Soundstream/Encodec is quite feasible. This integration is indeed the direction we plan to explore in our next steps.
> ### Q4: Scaling up model size
> We believe that as model size increases, its ability to learn effective representations will improve. Additionally, a larger model is likely to achieve a better balance between distillation and reconstruction tasks.

---

### Meta-Review · Area_Chair_hMd6 · 2023-12-01

**Metareview:**

A model, SpeechTokenizer , which uses a Encoder-Decoder architecture with residual vector quantization (RVQ)  to learn a speech tokenizer that can reconstruct speech  while retaining enough semantic information in speech is described in the paper. The proposed approaches overcomes some of the limitations of previous works leveraging separate tokenizers for semantic and acoustic information. The experimental findings appear to be still limited dispite the authors' effort in providing additional results but convincing. It is somehow original in the way the acoustic and sematic information is combined, yet the idea per is not totally new.  Assessing the proposed solution for speech understanding and speech generation is still missing, but overall the contribution is sound.

**Justification For Why Not Higher Score:**

The idea of combining acoustic and text based tokens is not fully new. Some experimental validation is  still needed.

**Justification For Why Not Lower Score:**

The results  could be useful for  advancement of research in  LLMs.

---

### Decision · Program_Chairs · 2024-01-16

Accept (poster)